# Vegetation structural change since 1981 significantly enhanced the terrestrial carbon sink

Jing M. Chen[1,2], Weimin Ju [2,3], Philippe Ciais [4], Nicolas Viovy[4], Ronggao Liu [5], Yang Liu [5] & Xuehe Lu [2]

Satellite observations show that leaf area index (LAI) has increased globally since 1981, but the impact of this vegetation structural change on the global terrestrial carbon cycle has not been systematically evaluated. Through process-based diagnostic ecosystem modeling, we find that the increase in LAI alone was responsible for 12.4% of the accumulated terrestrial carbon sink ($95 \pm 5$ Pg C) from 1981 to 2016, whereas other drivers of $CO_2$ fertilization, nitrogen deposition, and climate change (temperature, radiation, and precipitation) contributed to 47.0%, 1.1%, and −28.6% of the sink, respectively. The legacy effects of past changes in these drivers prior to 1981 are responsible for the remaining 65.5% of the accumulated sink from 1981 to 2016. These results refine the attribution of the land sink to the various drivers and would help constrain prognostic models that often have large uncertainties in simulating changes in vegetation and their impacts on the global carbon cycle.

[1] Department of Geography and Program in Planning, University of Toronto, Toronto, ON M5S 3G3, Canada. [2] Jiangsu Provincial Key Laboratory of Geographic Information Science and Technology, International Institute for Earth System Science, Nanjing University, Nanjing 210023, China. [3] Jiangsu Center for Collaborative Innovation in Geographic Information Resource Development and Application, Nanjing 210023, China. [4] Laboratoire des Sciences du Climat et de l'Environnement, LSCE/IPSL, CEA-CNRS-UVSQ, Universite Paris-Saclay, F-91191 Gif-sur-Yvette, France. [5] State Key Laboratory of Resources and Environmental Information System, Institute of Geographic Sciences and Natural Resources Research, Chinese Academy of Sciences, Beijing 100101, China. Correspondence and requests for materials should be addressed to W.J. (email: juweimin@nju.edu.cn)

Terrestrial ecosystems are an important part of the climate system and their effect on the global carbon cycle is one of the largest uncertainties in the projection of future climate[1,2]. The estimation of the terrestrial carbon cycle is complicated by large spatial and temporal variability of the vegetation cover, as well as complex biological, climate, and soil controls on plant growth and organic matter decomposition[3]. Because of this complexity, the estimates of the global terrestrial carbon sink using prognostic models differ considerably with the residual of the global carbon budget in terms of both the mean and inter-annual variations[4]. The attribution of the global land carbon sink to various drivers also differs greatly among models[5]. For the purpose of projecting future climate, we need to better understand the mechanisms controlling the terrestrial carbon cycle so that we can reliably estimate the terrestrial carbon cycle under future climatic and atmospheric conditions[6,7].

With human's perturbation to the climate system through greenhouse gas emissions to the atmosphere and land-use change, the terrestrial carbon cycle has been greatly altered since pre-industrial time[8,9]. Studies have shown that increased atmospheric $CO_2$ concentration since 1850 has enhanced plant growth and hence induced carbon sinks[10], although the magnitudes of this enhancement vary among model estimates, satellite-based assessments, and Free Air Enrichment Experiments performed at a limited number of sites[3]. Climate change and atmospheric nitrogen deposition also played important roles in modulating the terrestrial carbon sink[11,12]. In addition to direct effects of these drivers on the carbon cycle, they have also induced changes in vegetation structure, i.e., leaf area index (LAI), defined as one half the total leaf area per unit ground surface area[13–15], which in turn also changes the carbon cycle. This feedback of vegetation LAI to the carbon cycle has not yet been systematically studied, although the increase in LAI over the last several decades has been found to be significant and dubbed global greening.

Currently, prognostic models, which simulate vegetation structure, growth, and carbon cycle under given climatic and edaphic conditions, are used, e.g., by the Global Carbon Project (GCP) as the main tools to estimate the terrestrial carbon sink[4]. The simulated results vary among models due to different assumptions and parameter settings, causing uncertainties[3]. One of the largest variations among these models is the simulation of vegetation structural change with time. Without accurate assessment of this change, it is highly uncertain to attribute the land sink to the various drivers, even if the total land sink is adjusted to an appropriate range and constrained by measured atmospheric $CO_2$ concentration.

Reliable satellite measurements of LAI are available to assess vegetation changes at the global scale since 1981. This source of information is underutilized in constraining the estimation of the terrestrial carbon sink and closing the global carbon budget, although many studies showed the usefulness of LAI products in optimizing several ecosystem parameters[16]. To use this information beyond assessing vegetation greening/browning trend[14], diagnostic models that assimilate remotely sensed vegetation structural information to simulate physical, biological, and ecological processes in vegetation are effective tools to estimate the impact of LAI changes on the carbon cycle. In this study, we use a model of this type, which is named Boreal Ecosystem Productivity Simulator (BEPS)[17]. This model was initially developed for boreal ecosystems and has been adapted for all ecosystems over the globe[18]. BEPS mechanistically includes the impacts of various drivers on gross primary productivity (GPP) (climate, $CO_2$ concentration, and nitrogen deposition) and assimilates vegetation structure (LAI) satellite data. It differs from light-use efficiency (LUE) models, which estimate GPP based on radiation absorbed by the canopy and prescribed LUE functions that may or may not include $CO_2$ and nutrient effects[3]. BEPS also simulates the dynamics of carbon pools beyond GPP and uses a spin-up procedure to prescribe soil carbon pools for estimating autotrophic respiration (AR) and heterotrophic respiration (HR) (see Methods). It is therefore a diagnostic process model for estimating the full carbon cycle using remote-sensing data and suitable for ascribing land carbon sinks to the various drivers.

Based on three LAI time series derived from satellite data, we find that vegetation structural change reflected by the trend of LAI contributed 12.4% to the accumulated total terrestrial carbon sink ($95 \pm 5$ Pg C) from 1981 to 2016. This is small, but significant, compared with the contributions of $CO_2$ fertilization ($+47.0\%$) and climate change ($-28.6\%$) in the same period. This finding suggests the importance in tracking this vegetation structural parameter using satellite data in global carbon cycle research. Quantifying this separate effect of vegetation structural change on the land sink helps attribute the sink to the various drivers including $CO_2$ fertilization, climate change, and nitrogen deposition, and may also help rectify some differences among prognostic models.

## Results

**LAI data analysis.** A global LAI time series from 1981 to 2016 at 8 km resolution and 16-day (1981–2000) and 8-day (2001–2016) intervals was produced using Advanced Very High Resolution Radiometer (AVHRR) and Moderate Resolution Imaging Spectroradiometer (MODIS) satellite data with the GLOBMAP algorithm[19]. The global distribution of the temporal trend of LAI over this period is shown in Fig. 1. About 74.2% of the land surface shows an increasing trend, among which 52.7% is significant at $p = 0.05$ level (two-tailed). Globally, the increase in the maximum LAI in the peak growing season is about twice as large as that in the annual mean, suggesting that growing season lengthening is not the main factor explaining the greening trend. This general increase in LAI resulted from the combined effects of various drivers including $CO_2$, climate, and nitrogen deposition over the same period, and therefore provides a new base for separating the effects of these drivers on vegetation structure and growth. The focus of this study is on the increase of LAI on plant growth and the land carbon sink. Considering the uncertainty in the temporal trend of this LAI time series, we used two other LAI products: GLASS LAI[20] and GIMMS LAI3g[21] (Supplementary Fig. 1 and Table 1).

**Analysis of the residual land sink.** Driven by one of LAI time series at a time as well as climate, $CO_2$, soil, and nitrogen deposition data, BEPS is used to simulate GPP, AR, and HR at daily time intervals for each pixel. The sum of simulated net ecosystem productivity (NEP), taken as GPP-AR-HR, for all land areas is compared with the global residual land sink (RLS) reported by the GCP[4], which is computed as the sum of emissions from fossil fuel consumption, cement production, and land-use change minus the sum of $CO_2$ accumulated each year in the atmosphere and ocean. The modeled annual NEP as average from the BEPS model forced by each of the three LAI products closely follows the trend and interannual variability of the residual land carbon sink (Fig. 2), although it does not capture well extremely low and high values in some years, such as 1987, 1991, 2000, 2002, and 2009. Over the 1981–2016 period, the modeled accumulated NEP is $95 \pm 5$ Pg C, whereas the accumulated RLS is $94 \pm 5$ Pg C.

In comparison with 15 prognostic models used by GCP[4], BEPS is among the best in terms of Pearson's coefficient ($R^2$), root mean square error (RMSE) between simulated and the observation-based annual RLS, and the accumulated sink from 1981 to 2016 in comparison with RLS (Supplementary Table 2).

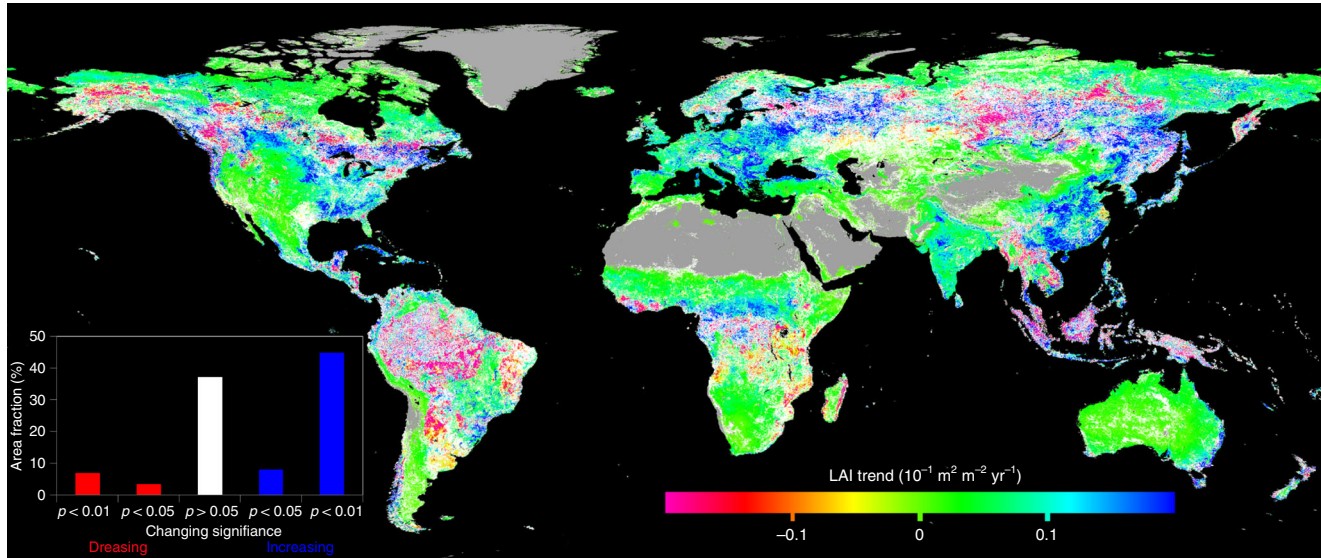

**Fig. 1** Global map indicating the trend of LAI from 1981 to 2016. The gray color indicates non-vegetated areas and the white color denotes that the trend is statistically insignificant ($p > 0.05$). Positive values indicate increasing trends of growing season mean LAI and vice versa

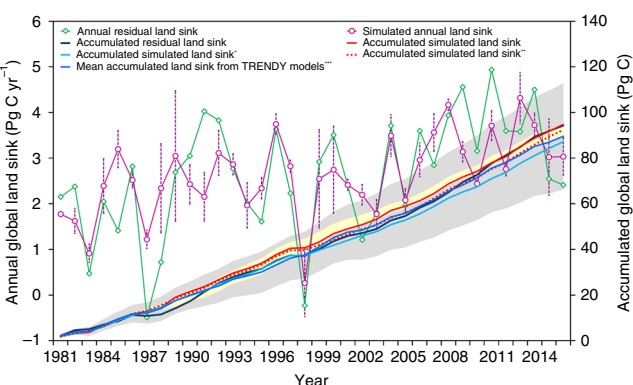

**Fig. 2** Comparison of the simulated annual land sink (NEP) by BEPS and the residual land sink (RLS) estimated by the Global Carbon Project. The pink red error bars are the SDs of the annual land sink simulated using three different LAI datasets. The solid red line indicates the accumulated carbon sink simulated using three LAI datasets. For the accumulated simulated land sink* (light solid blue line), pixels with >20% areal changes in short vegetation or tree canopy are excluded in the accumulation. For the accumulated simulated land sink** (dashed red line), pixels with >30% changes are excluded. The solid dark line indicates the accumulated residual land sink estimated by the Global Carbon Project. The solid blue line is the mean of accumulated land sinks simulated by 15 TRENDY models, and the shaded gray area represents its uncertainty (mean ± SD). The shaded light yellow area represents the range of accumulated land sinks simulated using three different LAI datasets

BEPS has an $R^2$ value of 0.56 (Supplementary Fig. 2), which is slightly lower than those of Community Atmosphere Biosphere Land Exchange (CABLE) and Lund–Potsdam–Jena (LPJ), but the RMSE of BEPS is lower than that of all prognostic models. The modeled accumulated sink by BEPS is also very close to the accumulated RLS, whereas estimates from the 15 prognostic models differ by a wide range. The results of these comparisons show that BEPS as a diagnostic model driven by remote-sensing data can do a similar or better job than fully prognostic models in simulating the past land sink, because vegetation structural changes observed by satellites provide a critical constraint to the sink estimation. We do not include other diagnostic models in these comparisons[22,23], because BEPS so far may be the only

process-based diagnostic model that calculates the full carbon cycle at the global scale for multiple decades. With the computation of the land sink close to the RLS, BEPS can then be used to attribute the sink to the various drivers including vegetation structural change.

The time-varying maps of satellite LAI constrain the effect of vegetation structural change on the land sink since 1981. However, the land sink in recent decades results from the accumulated changes in climate, $CO_2$ concentration, nitrogen deposition, and land use since the preindustrial period that occurred before 1981. From 1981 to 2016, global land ecosystems absorbed $95 \pm 5$ Pg C from the atmosphere (i.e., accumulated NEP), which is 32.8 Pg C larger than the baseline (62.1 Pg C) defined by NEP simulated assuming without changes of these drivers after 1981 (Supplementary Table 3). Several conditions were set for simulating the baseline. First, $CO_2$ concentration and nitrogen deposition were kept at the 1981 levels; second, LAI was taken as the mean value in 1982–1986; and third, meteorology in a year between 1981 and 2016 is randomly taken from meteorology in a year within the 1971–1979 period (Supplementary Table 3), so that no climate trend exists over the 1981–2016 period. Under the baseline conditions, the land sink over the period from 1981 to 2016 is thus caused only by the legacy of changes in the drivers prior to 1981, given the residence time of carbon in ecosystems. The mean legacy effect over 1981–2016 was larger in forests, especially in evergreen broadleaf forests over tropical regions (Supplementary Fig. 3). We therefore refer to the sink increase from the mean legacy effect as the sink enhancement due to changes of the drivers after 1981.

**Attribution of the land sink.** The increase in atmospheric $CO_2$ concentration is modeled to be the dominant diver for the land sink enhancement during the 1981–2016 period. The $CO_2$ concentration increase after 1981 alone enhanced the global land sink by 44.6 Pg C accounting for 47.0% of the total accumulated sink enhancement after 1981 (Fig. 3). The simulated global total net primary productivity (NPP) increased by 11.6% with the increase of $CO_2$ concentration from 340.13 p.p.m. in 1981 to 404.20 p.p.m. in 2016, whereas climate, LAI, and N deposition remain at baseline values. The sensitivity of simulated total NPP in the northern hemisphere to atmospheric $CO_2$ concentration ($\beta$-factor[3]) was 18.6%/100 p.p.m. during 1981–2016. This $\beta$-factor is

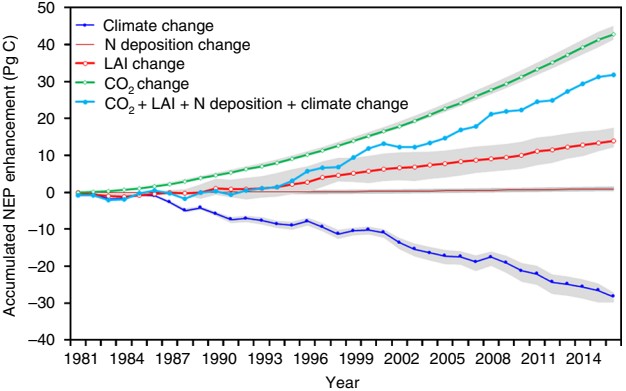

**Fig. 3** The accumulated terrestrial carbon sink from 1981 to 2016, after subtracting the baseline sink of 62.1 Pg C due to long-term changes (legacy effects). Through numerical experiments, the variations of the accumulated sink with time due to the various drivers are calculated, showing that $CO_2$ fertilization (without LAI changes), LAI, nitrogen deposition, and climate are responsible for the sinks of 44.6, 11.7, 1.1, and −27.1 Pg C over this period, respectively. These sinks contribute 47.0%, 12.4%, 1.1%, and −28.6% to the total accumulated residual land sink (95 ± 5 Pg C) over this period, respectively. For example, the accumulated $CO_2$ enhancement is calculated by holding other drivers at the baseline, while changing $CO_2$ according to the global mean $CO_2$ data

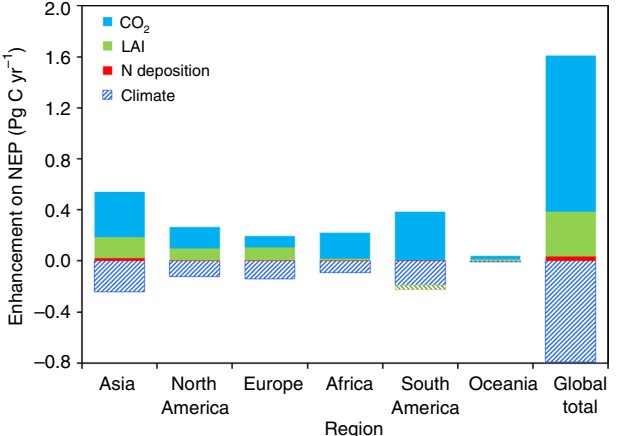

**Fig. 4** Attribution of the global land sink to various factors by region averaged over the period from 1981 to 2016. The values shown are regional and global totals of mean land sinks simulated with an individual diver ($CO_2$, LAI, nitrogen deposition, and climate) changing historically and other factors being remained temporally unchanged minus the lase sinks due to long-term changes (legacy effects)

about 6.1% smaller than the average value of prognostic models participating in the Coupled Model Intercomparison Project 5 (CMIP5)[3]. Our β-factor excludes the effects of $CO_2$ fertilization on LAI, whereas CMIP5 models lump vegetation LAI responses to $CO_2$ increase into the β-factor. If the influence of LAI change is included in our β-factor calculation, it would be 31.8%/100 p.p.m. for the northern hemisphere, whereas the global average value is 27.0%, because LAI increased much less in the southern hemisphere than in the northern hemisphere. Both BEPS and CMIP5 models include the influences of $CO_2$ increase on stomatal conductance affecting photosynthesis, transpiration, and soil moisture. Our β-factor is considerably larger than the values inferred from remote-sensing NPP models[3], because these models are mostly based on empirical LUEs that do not explicitly consider enhanced LUE at higher $CO_2$ levels[24] (see Supplementary Fig. 4

for further comparisons). $CO_2$ fertilization enhanced the carbon sink in all regions (Fig. 4), especially in regions with high NPP (15°S–30°N and 45°N–60°N) (Supplementary Fig. 5).

The impacts of other drivers on the accumulated sink shown in Fig. 3 are also calculated by changing one driver at a time, while holding other drivers at the baseline level. The accumulated global sink enhancement due to vegetation structure (LAI) change over the 1981–2016 period is 11.7 Pg C, which is 12.4% of the total sink or 35.7% of the enhanced sink in the same period. Over this period, global average LAI increased from 1.6 to 1.7, enhancing GPP by 1.2% and NEP by 0.3% relative to GPP. The spatial distribution of NEP enhancement (Supplementary Fig. 6) is similar to that of the LAI trend shown in Fig. 1, where positive trends of LAI induced sinks, whereas negative trends caused sources, suggesting that LAI might have acted as a surrogate for the impacts of changes in other factors such as soil moisture and temperature. As most areas show positive trends, the overall effect of the vegetation structural change is a large sink enhancement. Changes in atmospheric nitrogen deposition made global land ecosystems sequester 1.1 Pg C more carbon during the 1981–2016 period than the baseline (Fig. 3). Nitrogen deposition contributed to 1.1% of the total accumulated sink since 1981. Based on global nitrogen deposition data, global total nitrogen deposition increased from 42.3 Tg N per year in 1981 to 58.9 Tg N per year in 2016 (Supplementary Fig. 7), with a total additional cumulative input of 0.30 Pg N into land ecosystems above the 1981 baseline. Our simulated carbon sink enhancement per unit of deposited nitrogen is 3.7 g C/g N, which is slightly lower than the range of 4.3–4.8 g C/g N by previous global simulations[25,26]. The NEP enhancement during the 1981–2016 period by nitrogen deposition mainly occurred in Asia and in part of Europe, where nitrogen deposition continuously increased (Supplementary Fig. 8).

Globally, climate change weakened the land sink during the 1981–2016 period (Fig. 3), when it's effect on LAI, such as longer growing season, is excluded. Climate change induced an accumulated GPP reduction of 37.6 Pg C, whereas the accumulated decrease of ecosystem respiration was 10.5 Pg C during the 1981–2016 period. Consequently, the climate change caused a net reduction of 27.1 Pg C (−28.6%) in the accumulated sink enhancement since 1981. The decrease of the land sink due to climate change occurred almost in all regions (Fig. 4, Supplementary Fig. 9). The sum of the effects of changes in LAI, $CO_2$, nitrogen deposition, and climate mainly enhanced carbon sink in Eurasia, southeastern China, eastern North America, central Africa, and southeast Asia (Fig. 4, Supplementary Fig. 10). The dominant driver affecting the land sink varies spatially (Supplementary Fig. 11). Climate had the most dominant-negative impact on the accumulated carbon sink in 14.2% of the total vegetated area of the globe. LAI is the dominant-positive driver for 43.6% of the area, whereas it is negative for 4.6% of the area. $CO_2$ and nitrogen deposition are both positive, dominant factors over 36.4% and 0.2% of the area, respectively.

## Discussion

In principle, NEP from BEPS should not equal the RLS, because NEP excludes the net emission from anthropogenic and natural disturbances (land cover and land-use change, harvest, plantation, fires, and insects) (Supplementary Discussion). Those disturbances can induce both immediate LAI reduction and subsequent gradual LAI increase due to regrowth. In BEPS, a LAI reduction induces additional transfer of the same portions of biomass pools to soil organic matter, which is subsequently respired as a source of carbon to the atmosphere. This disturbance-enhanced carbon loss by respiration is mostly compensated by regrowth, which is driven by observed LAI series in

the same period, whereas NEP equals RLS for undisturbed areas. If the carbon gain through regrowth equaled the carbon loss due to enhanced respiration, the simulated NEP would equal RLS for disturbed areas. This would be true only if the disturbance and recovery were historically maintained at constant rates. However, there are considerable differences between NEP modeled by BEPS and RLS for areas with large changes in short vegetation (SV), tree cover (TC), and bare land. To assess these differences, we also show in Fig. 2 the simulated accumulated NEP curves after excluding pixels (8 km resolution) with changes in SV or TC coverage over 20% and 30%, respectively, in the accumulation, based on SV and TC high-resolution (30 m) map produced from Landsat data[27]. Excluding these areas with large SV or TC changes results in small decreases in accumulated NEP by 8.0% and 2.5% for these two cases, respectively, suggesting that cumulative NEP approximately equals cumulative RLS at the global scale since 1981. These effects of disturbance on simulated NEP are small, because pixels with 20% and 30% disturbance in SV or TC are only 6.7% and 2.0% of the land area, respectively (Supplementary Fig. 12), and the LAI observations used to drive BEPS can capture a large portion of the impacts of disturbance on photosynthesis and ecosystem respiration. These exclusions always cause smaller NEP values, because the contributions from undisturbed portions within the excluded pixels are discounted. Changes from SV to TC or from bare land to SV or TC within the excluded pixels are also discounted, although they may incur small sinks (Supplementary Fig. 13), which are associated with LAI changes (Supplementary Fig. 14). For simplicity, we do not exclude disturbed pixels in the results shown in Figs. 2 and 3.

In conclusion, the results of this study show the value of assimilating observed LAI over the last three decades to quantify the land carbon sink and separate the effect of increasing LAI alone vs. the effects of changes in environmental drivers alone. It should be kept in mind, however, that the observed increase of LAI also results from environmental drivers as well as of land use and land management.

## Methods

**GPP modeling methods.** The BEPS model[17,18] used in this study is a process-based diagnostic model driven by remotely sensed vegetation parameters, including LAI, clumping index, and land cover type, as well as meteorological and soil data. It simulates photosynthesis, energy balance, and hydrological and soil biogeochemical processes at daily time steps[17,28]. For GPP simulation, BEPS uses the leaf-level biochemical model[29] with a two-leaf upscaling scheme from leaf to canopy:[17]

$$GPP = GPP_{sun}LAI_{sun} + GPP_{shaded}LAI_{shaded} \qquad (1)$$

where $GPP_{sun}$ and $GPP_{shaded}$ are the GPP per unit area of sunlit and shaded leaves, respectively. $LAI_{sun}$ and $LAI_{shaded}$ are the LAI of sunlit and shaded leaves, respectively, and are estimated as:

$$LAI_{sun} = 2 \times \cos\theta \times \left[1 - \exp\left(-0.5 \times \Omega \times \frac{LAI}{\cos\theta}\right)\right] \qquad (2)$$

$$LAI_{shade} = LAI - LAI_{sun} \qquad (3)$$

where $\Omega$ is the clumping index derived from MODIS data at 500 m resolution[30] and $\theta$ is the daily mean solar zenith angle.

GPP values of sunlit and shaded leaves are calculated using the Farquhar's model[29] with consideration of the large difference in incident solar irradiance and the small difference in the carboxylation rate between these two-leaf groups[18]. Stomatal conductances of sunlit and shaded leaves are determined separately according to photosynthesis rates of these leaves, atmospheric $CO_2$ concentration, and soil moisture[28,31], and are used to estimate water consumption by evapotranspiration. Although initially developed to simulate GPP in boreal ecosystems in Canada, BEPS has been adopted and has been widely used to estimate terrestrial carbon and water fluxes in China[32,33], North America[34,35], Europe[36], East Asia[37], and the globe[18].

**Full carbon cycle modeling methods.** BEPS includes modules to calculate HR and NEP[28]. Based on a modified Century model[38,39], it stratifies the biomass carbon stock into four pools (leaf, stem, coarse root, and fine root pools) and the soil carbon stock into nine pools (surface structural litter, surface metabolic litter, soil

structural litter, soil metabolic litter, coarse woody litter, surface microbe, soil microbe, slow, and passive carbon pools). These carbon pools are initialized in the following way. The model first calculates NPP in 1901 using N deposition and $CO_2$ concentration in 1901, a random year of climate data selected in the period from 1901 to 1910, seasonally variable LAI averaged over the 1982–1986 period, and default C:N ratios for all carbon pools. The nine soil and four biomass carbon pools are then estimated under the assumption that the carbon cycle of terrestrial eco-systems was in dynamic equilibrium in 1901. With this assumption, all carbon pools are determined by solving a set of equations describing the dynamics of carbon pools[40]. For the period from 1901 to 1980, the model is run using historical data of N deposition, $CO_2$ concentration, and climate, and the average LAI during 1982 to 1986. Due to lack of data, we assume that LAI in 1982–1986 represents that in 1901–1981. If LAI increased in this period, NPP in 1901–1910 would be over-estimated, leading to larger soil carbon pools in 1901 and smaller NEP in 1981–2016. To address this issue, we conducted a set of simulations by extending the LAI time series to 1901 according to atmospheric $CO_2$ concentration with the rate of LAI change with $CO_2$ determined using 1981–2016 data. We find that the enhancement of the land sink due to LAI change in 1981–2016 decreases from 12.4% to 10.5% relative to the accumulated sink in the same period when the possible LAI increase from 1901 to 1981 is considered. This decrease is due to higher accumulated NEP by 6.8 Pg C during 1981–2016, resulting from lower initial soil carbon pools at 1901 when LAI is smaller than our simulations without considering LAI change over 1901–1981. This set of simulations suggests that the impact of possible LAI changes prior to 1981 on the role of LAI after 1981 is within a few percent and does not affect our conclusion on the significance of LAI increase after 1981 in enhancing the land sink. Although extrapolating LAI according to $CO_2$ concentration is overly simplistic, it may be considered as setting the upper bound of the possible error due to LAI changes prior to 1981, because climate change could have been negative on plant growth and LAI.

The decomposition of soil carbon and mineralization of soil nitrogen are regulated by soil temperature, moisture, texture, and chemical property of soil carbon pools. Nitrogen available for vegetation growth consists of the total of mineralized and deposited nitrogen. The uptake of nitrogen by vegetation is simulated according to temperature, total amounts of soil carbon and nitrogen, and vegetation demand. The absorbed nitrogen is allocated daily to different vegetation carbon pools based on the C:N ratios and NPP allocation coefficients. The nitrogen content of leaves is used to adjust the parameter Vcmax at 25 °C, which consequently affects the photosynthesis rates of sunlit and shade leaves in the Farghuar model[29]. AR consists of maintenance and growth respiration, and maintenance respiration depends on foliage, stem and root biomass, and temperature, whereas growth respiration is taken as a fraction (25%) of GPP. NEP of each modeling grid equals GPP-AR-HR.

**LAI data.** LAI is an input into the BEPS model for the simulation of the carbon flux. Three LAI time series, GLOBMAP-V2, GLASS, and LAI3g are used in this study and are shown in Supplementary Fig. 1 in comparison with other LAI time series. The GLOBMAP _V2 product is the basis for our simulations, whereas GLASS and LAI3g are used to assess uncertainties in carbon sink estimation due to the choice of LAI products. GLOBMAP_v2 over the period from 1981 to 2016 was generated through fusing LAI inverted from MODIS reflectance data with AVHRR GIMMS NDVI data. LAI from 2001 to 2016 was first derived from the MOD09A1 C6 land surface reflectance and the associated illumination and view angles based on the GLOBCARBON LAI algorithm[19,41], which was developed on the basis of the 4-Scale geometric optical model[42]. This algorithm explicitly considers the effects of the bidirectional reflectance distribution function on reflectance over the canopy as measured by the sensors[41]. For the fusion of MODIS and AVHRR remote-sensing data, the relationships between GIMMS NDVI and MODIS LAI were established by pixel over a period (2001–2006)[19] that they overlap. Then the AVHRR LAI from 1981 to 2000 was generated using these relationships, to ensure the temporal consistency between these two sensors. The spatial resolution of the LAI series is 0.072727° × 0.072727° and temporal resolution varies from 16 days (1981 to 2000) to 8 days (2001 to 2016). In the simulation, these 16 days and 8 days LAI values are interpolated into daily values. GLASS and LAI3g have the similar temporal coverage. All three long-term LAI series used in this study have similar magnitudes, because they have all considered the three-dimensional canopy structure, as characterized by the clumping index, in their retrieval algo-rithms[43]. Both GLOBMAP_V2 and GLASS used the same global clumping index map[30], whereas LAI3g considered clumping in a different way[21]. For accurate simulation of sunlit and shaded leaf area and GPP, both LAI and clumping index are needed.

All these three products used the processed AVHRR data (GIMMS). The issues with possible artifacts and errors in the AVHRR data series (GIMMS), such as sensor degradation, sensor intercalibration, orbital drift causing changes in sun-target-view geometry, distortions by clouds, and abnormal aerosol absorption by two major volcanic eruptions, have been fully considered and rectified to a large extent, ensuring the useful signals in the trend being extracted[44]. The GIMMS time series has quality flags with values from 0 to 6. GLOBMAP used only the top quality 0 or 1. Depending on the strength of quality control, different LAI products could show different interannual variations, although they are consistent in their increasing trends (Supplementary Table 1).

LAI data have been assimilated into diagnostic ecosystem models to optimize plant carbon allocation, stock, and residence time, as well as carbon-use efficiency[16,45]. In our study, we similarly optimized some of these parameters through a spin-up procedure and also used LAI data to calculate the long-term global carbon cycle.

**Meteorological data**. Meteorological data required to force the BEPS model include daily maximum and minimum temperatures, downward solar radiation, relative humidity, and precipitation. These data are interpolated from the $0.5° \times 0.5°$ CRUNCEP V8.0 dataset, which is a combination of CRU monthly climatology and 6 hourly NCEP reanalysis meteorological data[46]. Daily maximum and minimum temperatures, downward solar radiation, and precipitation are directly retrieved from the CRUNCEP V8.0 dataset. Relative humidity is calculated from temperatures, specific humidity, and pressure from the CRUNCEP V8.0 dataset. The $0.5° \times 0.5°$ meteorological data are interpolated into $0.072727° \times 0.072727°$ resolution using a bilinear interpolation method. These data are the same as those used by the TRENDY models.

**Soil data**. Fractions of clay, silt, and sand are retrieved from the harmonized global soil database (http://www.fao.org/nr/lman/abst/lman_080701_en.htm) and are used to determine soil physical parameters, including wilting point, field capacity, porosity, hydrological conductance, exponent of the moisture release equation, and so on.

**Nitrogen deposition data**. The yearly global nitrogen deposition data at $0.5° \times 0.5°$ resolution over the period from 1960 to 2009 are estimated from tropospheric $NO_2$ column density retrieved from Global Ozone Monitoring Experiment and Scanning Imaging Absorption Spectrometer for Atmospheric Cartography sensors, meteorological data, and NOx emission inventory data[47]. For the years from 2010 to 2016, nitrogen data are extrapolated using the estimated nitrogen data over the period from 2000 to 2009. For the period from 1901 to 1959, nitrogen data are extrapolated based on the change rates of nitrogen deposition over the period from 1960 to 1969. The $0.5° \times 0.5°$ nitrogen deposition data are interpolated into the $0.072727° \times 0.072727°$ resolution using a bilinear interpolation method.

The N deposition dataset used in this study is compared with that used by TRENDY models (Supplementary Fig. 2). These two datasets are similar before 1990, as both are based on measurements, but the increasing trends after 1990 are different, because the data used by the models are based on linear extrapolation from 1990 to 2050, at which the nitrogen deposition is estimated based on projected anthropogenic sources and other assumptions[48], whereas satellite measurements from 2000 to 2009 are used in our dataset[47] and could follow the realistic trend more closely than the linearly extrapolated trend used by the models. Over the 1981–2016 period, the total N deposition is 301 Tg N in our study, whereas it is 403 Tg N in TRENDY. The difference could be due to the overall decrease in N deposition in North America and other regions in this period.

**Uncertainty assessment**. The uncertainty of model results shown in Figs. 2 and 3 is estimated based on differences among simulated results using the three LAI products. This uncertainty is of a similar magnitude to those estimated from the uncertainties of model parameters that influence the simulated temporal trends, because bias errors are mostly constrained by the atmospheric $CO_2$ concentration and the main interest of this study is to attribute the land sink to the various drivers through their influences on the temporal trend. The uncertainty for the $CO_2$ fertilization effect, e.g., is mostly caused by the uncertainties in the slope and intercept of the linear relationship between stomatal conductance and photosynthesis rate. The trend of NEP against climate is mostly controlled by the sensitivities of AR and HR to temperature.

## Data availability
The global clumping data are available at http://globalmapping.org/CI/. The global GLOBMAPV3 LAI dataset for the period from 1981 to 2016 can be downloaded at http://globalmapping.org/globalLAI/.

## Code availability
The code used in this study is available from the corresponding author on request.

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

## Acknowledgements

This study is funded by the National Key R&D Program of China (2016YFA0600202) and in part by the National Natural Science Foundation of China (41671343) and by the Fundamental Research Funds for the Central Universities. P.C. acknowledges support from the European Research Council Synergy project SyG-2013-610028 IMBALANCE-P and the ANR CLAND Convergence Institute

## Author contributions

C.J.M. and J.W.M. designed this research and wrote the first draft of the manuscript. J.W.M. carried out all computations. C.P. helped with the assessment of disturbance effects and provided constructive comments on the manuscript and helped. V.N. assisted in the use of meteorological data and in writing part of the manuscript. L.R.G. and L.Y. generated the GLOMAP LAI dataset. L.X.H. provided the major part of the nitrogen deposition dataset.

## Additional information

**Competing interests:** The authors declare no competing interests.

