## [Peer Review File · Nature Communications]

Reviewers' comments:

Reviewer #1 (Remarks to the Author):

Nature Comms

Vegetation Structural Change Since 1981 Significantly Enhanced the Terrestrial Carbon Sink
Chen et al

Evaluating changes in the Terrestrial Carbon Sink is one of the most complicated tasks facing biogeoscientists because so many biophysical drivers are changing simultaneously on a global and the sensitivity to these drivers can vary by climate and ecological space. We know that CO₂ is increasing and the planet is warming. But superimposed on this secular trend are regions that are greening, others being lost to fire and deforestation. Increasing fossil fuel combustion is causing N deposition that may stimulate growth of some native ecosystems, but it also causes air pollution that may limit photosynthesis elsewhere.

This team focuses on the signal of a changing leaf area index. As a biophysical scientist and modeler, I appreciate that changing LAI may lead to changes in photosynthesis because the canopy carbon flux is proportional to LAI. But the response can be conditional, as a change in LAI at high values will lead to a smaller change in absorbed light and photosynthesis.

To make this point here is a simple computation of canopy photosynthesis as a function of LAI and light (see pdf)

The good thing is that the model is being applied globally to areas with high and low LAI, so it should be able to produce an appropriate globally integrated annual and decadal sum of the response of net carbon fluxes to LAI.

One question is how big may this sensitivity to LAI be?

We also have to consider integrating the fluxes over time and accumulate them over a year. In my past computations I have examined the sensitivities of annual fluxes of carbon dioxide to small differences in LAI. In one example I found that increasing LAI from 5 to 6 changes net ecosystem carbon exchange from -569 to -577 gC m⁻² y⁻¹, a 1.4% change. We have to remember that increasing LAI increases photosynthesis, but that respiration scales with photosynthesis, so the change in net carbon flux will be smaller. The point I am raising is that the change in LAI may be small and so may the annual sum of net carbon flux from a patch of land on a yearly basis.

In the work done here, the authors use the BEPS model. Among the suite of models used by the global carbon modeling community, I have higher confidence in BEPS than others. It does what is needed to be done to compute a suite of non linear functions correctly by upscaling leaf scale functions to that of the ecosystem. Most noteworthy it considers fluxes on the sun and shade fractions (Chen et al., 1999) and it considers clumping on light transfer through canopies. BEPS has been tested and validated with many exercises of direct eddy covariance data so I think the model is up for the task.

The lingering question is whether or not LAI is really changing with time or is the measure an artifact of sensor drift, pollution and other factors. Here we have to be careful as there are studies showing that trends in leaf area index may not be real (Jiang et al., 2017).

I find it ironic that this paper cites the Jiang et al paper, but it does not heed its lessons, Jiang et al conclude: 'Our results indicate that the four long-term LAI products were neither inconsistent over time nor interconsistent with each other. These inconsistencies may be due to NOAA satellite orbit changes and MODIS sensor degradation. Caution should be used in the interpretation of global

changes derived from the four long-term LAI products'.

So the paper under review becomes an example of simple plug and chug . Consequently, I cannot be confident its findings as true, especially with computed changes significant to 3 digits (12.4% of the accumulated carbon sink of 95 PgC over 1981 to 2016).

If the change in LAI is true, the authors do a nice job looking at other factors that I mentioned in the preamble. Figure 3 for example has computations with changes in N deposition, CO₂, warming and LAI. This helps bound the problem, but many other model papers have done similar computations, so can this be considered new and significant.

I do appreciate the effort to try and tease out LAI as a co driver to net carbon fluxes and think the model is appropriate. I just don't have the faith in the quality of the inputs of LAI, given the high inconsistency in trends of the different LAI products. The authors need to consider and discuss this inconsistency more and better if they aim to get this work published.

Chen, J.M., Liu, J., Cihlar, J. and Goulden, M.L., 1999. Daily canopy photosynthesis model through temporal and spatial scaling for remote sensing applications. *Ecol. Model.*, 124(2-3): 99-119.
Jiang, C. et al., 2017. Inconsistencies of interannual variability and trends in long-term satellite leaf area index products. *Global Change Biology*: n/a-n/a.

Reviewer #2 (Remarks to the Author):

Review of "Vegetation Structural Change Since 1981 Significantly Enhanced the Terrestrial Carbon Sink" by J. Chen et al.

this is a clearly written manuscript addressing the important issue of how vegetation structure (in this case primarily meaning changes in remote sensed leaf area index) over the satellite period has changed land carbon uptake. The paper concludes that the increasing trend in LAI plays a significant role (albeit smaller than CO₂) in changes in the terrestrial carbon sink. The paper also makes a nice split in the changes in carbon sink due to legacy effects of drivers prior to 1981 persisting versus changes after 1981.

I found the paper interesting and relevant to a Nature readership. The analysis is clearly laid out and appears robust and convincing. My only concern is that I found it a bit confusing which aspects of carbon changes were included/excluded or where. I outline below what I think you are saying and make suggestions how to improve the clarity of this coming across. Remember that you are very closely involved in the work, but a reader seeing it for the first time does not immediately grasp exactly the choices you have made for where to include certain terms.

if this can be clarified then I would find the paper acceptable for publication.
Chris Jones.

Major Comments

On first reading I was confused over how you treated land-use/change. It seemed both that some of your signal of LAI was driven by land-use change, but also you tried to exclude some of it. On closer re-reading this made me wonder how various aspects/drivers of carbon change were included in your separated terms.

You try to go beyond a traditional split into "climate" vs "CO₂" drivers which is what would be

gained for example from the TRENDY simulations. This is good. Based on that as a starting point such modelling studies would include the following processes/responses:

CO2 effects:

- 1a. CO2 induced additional growth ("CO2 fertilization")
- 1b. enhanced growth due to improved water use efficiency
- 1c. enhanced growth due to increased leaf area as a result of higher CO2

Climate effects:

- 2a. changes in vegetation productivity in a different climate
- 2b. changes in carbon residence time (esp. in soil) in a different climate
- 2c. changes in vegetation growth due to leaf area increase in a warmer climate

land-use:

3. changes in vegetation growth and carbon store due to land use change

In your analysis you split these in a different manner so that:

CO2 - is now made up of 1a + 1b from above. But you exclude 1c.

Climate - is now made up of 2a + 2b from above but you exclude 2c.

LAI - now you have a new term, made up from 1c + 2c above - which is good

BUT: the role of land use is not certain. SOME of it is included in your LAI term because you pick it up as a remote-sensed change in LAI. But SOME of it is excluded if it exceeds your (arbitrary) threshold of 20/30%. SO in addition to 1c + 2c you partially include a land-use term in your LAI sink.

I hope this is a correct reading of your results. If not then it highlights that it was not clear to follow. But if it is correct then I think you need to explain why splitting the land-use term partly into the LAI sink and partly excluded makes sense.

Other major comments

1. You say that BEPS may be the "only process-based diagnostic model that calculates the full carbon cycle". I'm not sure this is true, although it might depend on the subtlety of how you define "diagnostic process based". I think your `_usage_` of such a model is novel, but there are other studies with model-data fusion approaches which mix processes and data-forcing. E.g. I missed any reference to the Bloom et al. (PNAS, 2016) study using the CARDAMOM system. They take a somewhat similar approach to you in terms of using a simple model, but data driven, to quantify recent carbon sinks. They also include LAI as a driving data source, although they don't split their results into the single components. There's a nice review of EO data constraints on carbon cycle modelling by Exbrayat et al (Surveys in Geophysics, 2019, <https://doi.org/10.1007/s10712-019-09506-2>)

2. I like the split into forcing prior to 1981 and that of changes since. I also found that N-dep forcing since 1981 had contributed virtually nothing - this in itself is worthy of being more prominent (maybe that's another study). Many would be surprised by this important result.

3. It would be interesting to have at least a short description on the sub-annual characteristics of the LAI changes. Are you seeing changes in the peak, or just a lengthening of the growing season? can you say which is more important at driving a carbon sink? There is much literature on this and the offset of increased respiration too. The recent review by Piao et al (GCB, 2019) has a lot of detail. Also, e.g. Buermann et al (Nature, 2018).

4. in places I felt the methods could be described in more detail in the main text. I don't know how to recommend how to achieve this within word count, but just to suggest you describe the spin-up process. After reading the main text I was fairly sceptical about the nature of how you represent the sink prior to and after 1981. But when I read the supplement this became clear and I am happy with the method. But a reader might also query this - I would suggest at least a mention that you initialise your carbon pools with a pre-industrial spin up and then run up to 1981 before starting your factorial simulations from there.

5. related to this I question your assertion that you don't include any LAI changes during the period up to 1981. Given that CO₂ has increased by about the same amount before/after 1981 (approx. 280-340 and then 340-400 ppm), then naively I might expect the LAI change up to 1981 is similar magnitude to the change after it. Have you looked at the sensitivity of your results to the assumption it did not change? What if you extrapolated back to pre-industrial a LAI change of equal magnitude to the changes after 1981? How would this affect your legacy sink?

Minor comments

1. Abstract. line 47. Can you explain what you mean by "positive" feedback? Do you mean positive as in "good" in a subjective sense? or positive as in "an amplifying feedback"? If the latter, then I don't understand - you show that LAI changes have caused a carbon sink - surely this is a negative feedback?

2. you describe your model as "diagnostic" - I'm not sure what this means exactly. I would say at least parts of it are prognostic. It is driven by observations and forced by the observed LAI - that's fine. But in the sense that it has a memory (i.e. "state variables") which evolve from timestep to timestep, then it is a prognostic model. Certainly, the soil component, if based on CENTURY, is very similar in structure to the other process based prognostic models you describe.

3. lines 233/234. You don't need to show working out to calculate a percentage.

4. SI section 2.2. Can you briefly describe how this N-dep data set compares with that used within CMIP6 and TRENDY for driving process models?

5. Suppl. figures 3, 4 etc. It would be useful to be able to relate the threshold inland-use of 20/30% to your x-axis here, which is expressed in % per year. Perhaps you could mark a dashed line upwards from the points you include/exclude due to land-use

6. Suppl. figures 3-5. Please can you use a common colour scale with zero in the same place? In each of these figures, the zero point is different which means sometimes yellow is a positive value, sometimes negative etc. It would make them more readable as a set if the same colours represented positive/negative changes.

7. Suppl. methods lines 193-196 - please check figure numbering here. Do you really mean figures 5 and 6?

8. Suppl. figures 7 onwards - the maps of different runs. Please check the caption definition of which simulations you show and check against the table listing the runs. E.g. fig S7 says "Simulation VI" but I think it is "II" in the table? S8 says "II" but means "III"?

9. these maps are really nice - please can you add the final one showing the committed sink just from the changes up to 1981. You conclude that this is about 60% of the total carbon sink, so it would be nice to see the map too.

Reviewer #3 (Remarks to the Author):

This paper addressed an important question regarding the impact of long term vegetation changes on the Terrestrial Carbon Sink, using remote sensing data from 1981 to 2016. To my knowledge, the work is original. I judged the article and supplementary material well-structured and scientifically high level. I have some comments, mostly related to vegetation remote sensing, my domain of expertise.

Major comments:

The long term trend of LAI have been studied by Jiang et al. (2017). The current article relies on these previous study to analyze the impact of LAI trend to GPP and carbon sink. Jiang et al. clearly concluded (section 4.3) by: "A single product or even ensemble of the four products may not be directly considered as relevant references when interpreting long-term global carbon and water cycle studies during pre-MODIS period (1982 – 1999), except for several consensus: (i) trend signs, (ii) interannual variability values in high latitude regions, and (iii) annual anomalies in arid regions.". However, the current article is based on the trend on LAI, not only the sign (see "(i)" above) but also the value of the trend I think the authors should demonstrate the validity of using trend values knowing the conclusion of Jiang et al.

Can we trust the LAI variation, especially when we know the poor quality of the LAI products for pure-AVHRR-era (<2000)? I understand that the attribution of LAI to global land sink (figure 4) is determined by the analysis of Simulation III vs Simulation I. Therefore authors compare Simulation with LAI from 1982-86 (Simulation I) and Simulation with actual Changing LAI values (1981-2016) (Simulation III). However, the Supplementary Figure 1 (from Jiang et al. 2017 – original copied below) highlights the fact that there is a significant change for all product from 1999 (pure-AVHRR-era) to 2003 (MODIS-era). Therefore what is the validity of the comparison between Simulation I and Simulation III?

[please insert attachment_1 here]

The trend is derived from one LAI product and two more are analyzed in supplementary material. I suggest to better argue why they choose GLOMAP product as core, and GLASS and GIMMS-LAI3g, as secondary. Why not using all products covering the entire period? Indeed Jiang et al. concluded that TCDR was the most intraconsistent product after GLOMAP. Why removing TCDR from the analysis in the supplementary material?

Minor comments:

Figure 1: describe the lower left graph. In order to improve readiness of the graph, I suggest to better identify "increase" and decrease" in the xlabel.

Figure 1 (and many other places): what does the unit "a-1" refers to? Is there a confusion with year (yr-1)? If so homogenize.

L. 174: for more clarity, I suggest to add, like in supplementary (section 4), "Disturbance (i. e. land cover change)"

Figure 4: Not very clear - I understand that positive and negative values were separated to draw the cumulative bar lines. I would suggest to better distinguish positive and negative values (see graphical suggestion)

[please insert attachment_2 here]

Supplementary Figures 7-10: these figures are informative and map of results would be appropriate at the end of the core article. However, I understand that the 4 full map could not be placed in the article. I would suggest several options (but authors may think to better ideas):

- add fig 8 in article since LAI is the main variable of interest
- add maps of relative contribution of dNEP (but authors have to deal with positive vs negative contributions)
- use a RGB color composite by mapping a set of three most important contributions

Typographical mistakes:

- L.84: "total lank sink" => "total land sink"
- Supplementary Table 1: Row3, Col 1: "I" => "II"

Reviewers' comments:

Reviewer #1 (Remarks to the Author):

Nature Comms

Vegetation Structural Change Since 1981 Significantly Enhanced the Terrestrial Carbon Sink
Chen et al

Evaluating changes in the Terrestrial Carbon Sink is one of the most complicated tasks facing biogeoscientists because so many biophysical drivers are changing simultaneously on a global and the sensitivity to these drivers can vary by climate and ecological space. We know that CO₂ is increasing and the planet is warming. But superimposed on this secular trend are regions that are greening, others being lost to fire and deforestation. Increasing fossil fuel combustion is causing N deposition that may stimulate growth of some native ecosystems, but it also causes air pollution that may limit photosynthesis elsewhere.

Thank you for seeing the relevance of our work.

This team focuses on the signal of a changing leaf area index. As a biophysical scientist and modeler, I appreciate that changing LAI may lead to changes in photosynthesis because the canopy carbon flux is proportional to LAI. But the response can be conditional, as a change in LAI at high values will lead to a smaller change in absorbed light and photosynthesis.

To make this point here is a simple computation of canopy photosynthesis as a function of LAI and light (see pdf)

The good thing is that the model is being applied globally to areas with high and low LAI, so it should be able to produce an appropriate globally integrated annual and decadal sum of the response of net carbon fluxes to LAI.

We appreciate this insightful comment. In fact, this is one of the reasons that we did modeling even though global greening as indicated by the increase in LAI has already been published. As you pointed out, the carbon uptake and LAI are not linearly related to each other, so the right way to do it is to consider how LAI changes the radiation absorption in an ecosystem model, which is exactly what we did. Simply linearly relating GPP to LAI would cause the problems you pointed out. Even a nonlinear relationship between GPP and LAI, such as the use of light use efficiency model, would also be problematic because of the complex interactions among many drivers.

One question is how big may this sensitivity to LAI be?

We also have to consider integrating the fluxes over time and accumulate them over a year. In my past computations I have examined the sensitivities of annual fluxes of carbon dioxide to small differences in LAI. In one example I found that increasing LAI from 5 to 6 changes net ecosystem carbon exchange from -569 to -577 gC m⁻² y⁻¹, a 1.4% change. We have to remember that increasing LAI increases photosynthesis, but that respiration scales with photosynthesis, so the change in net carbon flux will be smaller. The point I am raising is that the change in LAI may be small and so may the annual sum of net carbon flux from a patch of land on a yearly basis.

For areas with large LAI, a small increase in LAI could indeed induce a very small increase in GPP, and the increase in NEP on the annual basis would even be smaller because of the increase in respiration due to warming. These processes are all considered in our model. At the global scale, we are talking about a very small increase in NEP due to LAI increase, which is about 2.93 gC m⁻²y⁻¹ or 0.3% of GPP on average. To answer your question quantitatively, we reanalyzed our model outputs and found that for pixels (59% of vegetated area) with positive LAI trends of 0-0.01 per year, the GPP increase due to LAI change over 1981-2016 is 6.5% at LAI<1 and 0.2% at LAI>3. For pixels (12%) with trends of 0.01-0.02 per year, the corresponding values are 8.8% and 0.8%. These statistics confirm your point that for the same LAI increase the effect on GPP is smaller at higher LAI. Since the global average LAI is 1.7 and the significant LAI trends fall in the range from 0 to 0.02 per year, the global average GPP enhancement due to LAI increase over 1981-2016 is 1.2%. To reflect this point, the following sentence is added in Lines 244-245: “Over this period, global average LAI increased from 1.6 to 1.7, enhancing GPP by 1.2% and NEP by 0.3% relative to GPP”

In the work done here, the authors use the BEPS model. Among the suite of models used by the global carbon modeling community, I have higher confidence in BEPS than others. It does what is needed to be done to compute a suite of non linear functions correctly by upscaling leaf scale functions to that of the ecosystem. Most noteworthy it considers fluxes on the sun and shade fractions (Chen et al., 1999) and it considers clumping on light transfer through canopies. BEPS has been tested and validated with many exercise of direct eddy covariance data so I think the model is up for the task.

Thank you for your knowledgeable and positive comments on our model. In fact, without an elaborate radiative transfer model which considers the three-dimensional canopy structure through the clumping index and LAI and separates the canopy into sunlit and shaded fractions, the influence of LAI on GPP would not be accurately quantified.

The lingering question is whether or not LAI is really changing with time or is the measure an artifact of sensor drift, pollution and other factors. Here we have to be careful as there are studies showing that trends in leaf area index may not be real (Jiang et al., 2017).

I find it ironic that this paper cites the Jiang et al paper, but it does not heed its lessons, Jiang et al conclude: ‘Our results indicate that the four long-term LAI products were neither intraconsistent over time nor interconsistent with each other. These inconsistencies may be due to NOAA satellite orbit changes and MODIS sensor degradation. Caution should be used in the interpretation of global changes derived from the four long-term LAI products’.

We have been concerned that Jiang’s paper could potentially cripple our paper because it paints a rather gloomy picture on the quality of LAI products, and it was our fault that we did not provide a sufficient assessment of the quality of LAI data used in our study. As a scientist, who worked on LAI measurement and mapping for a long career (with total citations in exceed of 24K), the first author Jing Chen fully appreciates the challenges and issues with global LAI products, but he also has confidence on the long-term increasing trend of the selected LAI products. First, six out of eight LAI products included in Jiang’s paper show increasing trends either for the long term since 1981 or for the short term after 2000, and the two products showing negative trends are both due to a data processing issue of MODIS C5. The MODIS C6 product after correcting for the issue due to sensor degradation shows an increasing trend, consistent with AVHRR-based products. The second product GLOBMAP showing a decreasing trend because it uses MODIS C5 reflectance data. After using MODIS C6 data, GLOBMAP_V2 (shown in supplementary Figure S1) also shows an increasing trend. Therefore, all corrected LAI products show increasing trends. The second issue is the interannual variability being considerably different among the long-term products, with GLOBMAP_V2 having the smallest variability and GLASS having the largest variability. GLOBMAP_V2, uses an algorithm developed by Jing Chen’s group (Deng et al., 2006), made a large effort in improving the intraconsistency of its time series using both AVHRR and MODIS data. A pixel-based reflectance matching procedure over a six-year overlapping period (2001-2006) was implemented. This procedure ensured a smooth transition from AVHRR to MODIS. In addition, a subpixel cloud removal and seasonal trajectory smoothing algorithm LACC (Locally Adjusted Cubic-spline Capping, Chen et al., 2006) is implemented and greatly reduced interannual variability of LAI. The GLASS product is also subsequently improved after Jiang’s paper and its interannual variability is much reduced (see Figure 1 below). The difference in the absolute value of LAI is not a serious issue in our study. MODIS C6 has smaller LAI values because foliage clumping is not fully considered (see review by Chen 2018). Both GLOPMAP and GLASS uses a global clumping index map (Chen et al., 2005; He et al., 2012) and therefore their LAI magnitudes are similar. LAI3g also considers the effect of clumping (Chen, 2018), so its LAI magnitudes are similar to those of GLOPMAP and GLASS. MODIS does not fully consider the effect of foliage clumping (Chen, 2018), showing smaller LAI values. For accurate simulation of sunlit and shaded leaf area, both LAI and clumping need to be considered.

Table A1. Trends of global annual mean LAI ($m^2 m^{-2} a^{-1}$) of different LAI products

Data source	Trends of global annual mean LAI						
	GLOBMAP 1982-2016	GLASS 1982-2015	TCDR 1982-2015	LAI3g 1982-2011	GLOBMAP 2001-2016	MODIS 2001-2016	GEOV1 1999-2013
AVHRR+ MODIS	0.0031^{***} (0.00039)						
AVHRR		0.0042^{***} (0.00063)	0.0052^{***} (0.00062)	0.0049^{**} (0.00108)			
MODIS					0.0068^{***} (0.00078)	0.0052^{***} (0.00074)	
SPOT/VGT							0.0154^{***} (0.0022)

Note: Values in brackets are uncertainties; *** and ** indicate the significance levels of 0.001 and 0.01,

respectively.

Figure A1. Comparison of latest LAI products (trend and uncertainty are given in Table A1)

The issues with possible artifacts and errors in the AVHRR data series (GIMMS), such as sensor degradation, sensor intercalibration, orbital drift causing changes in sun-target-view geometry, distortions by clouds, and abnormal aerosol absorption by two major volcanic eruptions, have already been fully considered and rectified to a large extent, ensuring the useful signals in the trend being extracted (Tucker et al., 2005). The GIMMS time series are used by all LAI products used in our study. It has quality flags with values from 0 to 6. The GLOMAP used only the top quality 0 or 1. Depending on the strength of quality control, different LAI products could show different interannual variabilities, although they are consistent in their increasing trend (Table A1).

Figure A1 and Table A1 are now provided in the supplementary material, so are the major points given above (Lines 84-99).

Chen, J. M., F. Deng, and M. Chen, 2006. Locally-Adjusted Cubic-Spline Capping Method for Reconstructing Seasonal Trajectories of a Surface Parameter Derived from Remote Sensing. *IEEE Transactions of Geoscience and Remote Sensing*, 44: 2230-2238

Chen, J. M., 2018. Remote Sensing of Leaf Area Index and Clumping Index. Chapter in *Comprehensive Remote Sensing*, Volume 3, edited by Shunlin Liang, Elsevier, Oxford, Pages 53-77, ISBN 9780128032213, DOI: 10.1016/B978-0-12-409548-9.10540-8.

Tucker, C. J., J. E. Pinzon, M. E. Brown, D. A. Slayback, E. W. Pak, R. Mahoney, E. F. Vermote, and N. El Saleous (2005), An extended AVHRR 8-km NDVI dataset compatible with MODIS and SPOT vegetation NDVI data, *Int. J. Remote Sens.*, 26(20), 4485–4498, doi:10.1080/01431160500168686.

(Other cited references are found in supplementary material).

So the paper under review becomes an example of simple plug and chug. Consequently, I cannot be

confident its findings as true, especially with computed changes significant to 3 digits (12.4% of the accumulated carbon sink of 95 Pg C over 1981 to 2016).

With the selected LAI products based on completeness of physics and data processing, the uncertainty in our simulated NEP trend is much smaller than that of the residual land sink (Figure 2), making the sink attribution to LAI change reliable. Although our work may appear to be a “simple plug and chug”, it actually involves quite a bit of pains-taking consideration of first order physics, including the differences among plant functional types in canopy architecture with the use of clumping index, areal fractions of various PFTs in a pixel, spinning up of soil carbon pools by PFT, etc. These considerations ensured the results to be reliable.

If the change in LAI is true, the authors do a nice job looking at other factors that I mentioned in the preamble. Figure 3 for example has computations with changes in N deposition, CO₂, warming and LAI. This helps bound the problem, but many other model papers have done similar computations, so can this be considered new and significant.

I do appreciate the effort to try and tease out LAI as a co driver to net carbon fluxes and think the model is appropriate. I just don't have the faith in the quality of the inputs of LAI, given the high inconsistency in trends of the different LAI products. The authors need to consider and discuss this inconsistency more and better if they aim to get this work published.

We thank you for making these positive comments. This is particularly much appreciated because indeed, we have made much effort in conducting this comprehensive modeling study after years of data preparation. We hope that with the explanation of the issues with LAI products (see above), you would have the same confidence as us in our conclusions.

Chen, J.M., Liu, J., Cihlar, J. and Goulden, M.L., 1999. Daily canopy photosynthesis model through temporal and spatial scaling for remote sensing applications. *Ecol. Model.*, 124(2-3): 99-119.

Jiang, C. et al., 2017. Inconsistencies of interannual variability and trends in long-term satellite leaf area index products. *Global Change Biology*: n/a-n/a.

Reviewer #2 (Remarks to the Author):

Review of "Vegetation Structural Change Since 1981 Significantly Enhanced the Terrestrial Carbon Sink" by J. Chen et al.

this is a clearly written manuscript addressing the important issue of how vegetation structure (in this case primarily meaning changes in remote sensed leaf area index) over the satellite period has changed land carbon uptake. The paper concludes that the increasing trend in LAI plays a significant role (albeit smaller than CO₂) in changes in the terrestrial carbon sink. The paper also makes a nice split in the changes in carbon sink due to legacy effects of drivers prior to 1981 persisting versus changes after 1981.

I found the paper interesting and relevant to a Nature readership. The analysis is clearly laid out and

appears robust and convincing. My only concern is that I found it a bit confusing which aspects of carbon changes were included/excluded or where. I outline below what I think you are saying and make suggestions how to improve the clarity of this coming across. Remember that you are very closely involved in the work, but a reader seeing it for the first time does not immediately grasp exactly the choices you have made for where to include certain terms.

if this can be clarified then I would find the paper acceptable for publication.

Chris Jones.

Thank you for your positive comments and insightful evaluation. It has been a challenge for us to deliver the complex issues in simple terms. Your suggestions are very helpful.

Major Comments

On first reading I was confused over how you treated land-use/change. It seemed both that some of your signal of LAI was driven by land-use change, but also you tried to exclude some of it. On closer re-reading this made me wonder how various aspects/drivers of carbon change were included in your separated terms.

Yes, land cover/use changes are part of the processes influencing LAI change. However, they are not the focus of this study. We excluded pixels with large disturbance fractions and showed that land use/cover changes did not alter significantly global NEP calculated based on LAI without changing land cover types. This is because the total disturbed fraction is small relative to the total land surface area and the disturbance rate does not change greatly so that regrowth is approximately balanced by the enhanced respiration after disturbance. We revised our manuscript to make this point clear in Lines 208-209: “*For simplicity, we do not exclude disturbed pixels in subsequent results shown in Figures 3 and 4*”.

You try to go beyond a traditional split into "climate" vs "CO₂" drivers which is what would be gained for example from the TRENDY simulations. This is good. Based on that as a starting point such modelling studies would include the following processes/responses:

CO₂ effects:

- 1a. CO₂ induced additional growth ("CO₂ fertilization")
- 1b. enhanced growth due to improved water use efficiency
- 1c. enhanced growth due to increased leaf area as a result of higher CO₂

Climate effects:

- 2a. changes in vegetation productivity in a different climate
- 2b. changes in carbon residence time (esp. in soil) in a different climate
- 2c. changes in vegetation growth due to leaf area increase in a warmer climate

land-use:

3. changes in vegetation growth and carbon store due to land use change

In your analysis you split these in a different manner so that:
CO₂ - is now made up of 1a + 1b from above. But you exclude 1c.
Climate - is now made up of 2a + 2b from above but you exclude 2c.
LAI - now you have a new term, made up from 1c + 2C above - which is good

BUT: the role of land use is not certain. SOME of it is included in your LAI term because you pick it up as a remote-sensed change in LAI. But SOME of it is excluded if it exceeds your (arbitrary) threshold of 20/30%. SO in addition to 1c + 2c you partially include a land-use term in your LAI sink.

I hope this is a correct reading of your results. If not then it highlights that it was not clear to follow. But if it is correct then I think you need to explain why splitting the land-use term partly into the LAI sink and partly excluded makes sense.

Your dissect of the various processes as shown about is correct and very helpful. Since we have LAI at hand, the impact of changes in LAI due to all factors on the carbon cycle is considered separately from other drivers, and therefore the terms 1c and 2c are indeed separated from the CO₂ and climate drivers. These separations are essentially the main contributions from this paper, as it helps to understand the land carbon cycle and to identify the directions to improve the global carbon cycle estimation. As the focus of this study is on the effect of LAI on the land sink, we don't intend to quantify the influence of disturbance on the sink, but rather we just want to ensure that historical disturbance does not significantly influence our conclusion on the LAI effect. This is done by excluding pixels with large disturbed areal fractions (20% and 30%), and we also showed in Figure S4 that there are very few pixels with disturbed fractions exceeding 30%. In other words, although there are complications in completely excluding the disturbance effects, the influence of disturbance on the effect of LAI change on the land sink is insignificant. We added the following two sentences at Lines 113-117: *"This general increase in LAI resulted from the combined effects of various drivers including CO₂, climate and nitrogen deposition over the same period, and therefore provides a new base for separating the effects of these drivers on vegetation structure and growth. The focus of this study is on the increase of LAI on plant growth and the land sink."*

Other major comments

1. You say that BEPS may be the "only process-based diagnostic model that calculates the full carbon cycle". I'm not sure this is true, although it might depend on the subtlety of how you define "diagnostic process based". I think your usage of such a model is novel, but there are other studies with model-data fusion approaches which mix processes and data-forcing. E.g. I missed any reference to the Bloom et al. (PNAS, 2016) study using the CARDAMOM system. They take a somewhat similar approach to you in terms of using a simple model, but data driven, to quantify recent carbon sinks. They also include LAI as a driving data source, although they don't split their results into the single components. There's a nice review of EO data constraints on carbon cycle modelling by Exbrayat et al (Surveys in Geophysics, 2019, <https://doi.org/10.1007/s10712-019-09506-2>).

There is a similar model named BESS (Breathing Earth Simulation System) by Ryu et al. (2011). However, it is mostly for GPP and ET calculations, and has not been applied to the historical

long-term LAI data (at least at the time this paper was written) for the full carbon cycle estimation. Thank you for providing the references for data assimilation works. Bloom et al. (2016) did a nice work in assimilating LAI and biomass data in a diagnostic model to optimize plant carbon allocation, stock, and residence time as well as carbon use efficiency, showing usefulness of LAI products for these purposes. Exbrayat et al. (2019) provided an updated review on the use of space data for improving our understanding of the land carbon cycle. In the review, LAI is assimilated into ecosystem models to improve various parameters and into atmospheric inversions models to improve land carbon flux inversion. It is also clear from this review that the potential in using LAI is diverse. After reading these references, we are more convinced that for the purpose of isolating the effect of LAI change on the terrestrial carbon cycle, a diagnostic modeling approach as shown in our paper is an effective way. Assimilating the long time series of LAI into a prognostic model is something yet to be done and but it requires much greater effort. We have referred to Exbrayat et al. (2019) in the paper (Lines 87-89) and added this discussion in the supplementary material (Lines 100-104).

Ryu, Y., D. Baldocchi et al., 2011, Integration of MODIS land and atmosphere products with a coupled-process model to estimate gross primary productivity and evapotranspiration from 1 km to global scales. Global Biogeochemical Cycles, VOL. 25, GB4017, doi:10.1029/2011GB004053

2. I like the split into forcing prior to 1981 and that of changes since. I also found that N-dep forcing since 1981 had contributed virtually nothing - this in itself is worthy of being more prominent (maybe that's another study). Many would be surprised by this important result.

We agree with your observation of this result from our work. There is no enough space to analyze this part of our work, but the likely reason is that the nitrogen deposition rate has been decreasing in some part of the world and new inputs of nitrogen could barely keep up with the increased demand of growth due to higher LAI. Globally, the nitrogen deposition rate was 42.3 Tg N yr⁻¹ in 1981 and increased to 58.9 Tg N yr⁻¹ in 2016. The total additional cumulative nitrogen input into land ecosystems over the 1981-2016 period is 0.30 Pg N above the 1981 baseline. Our simulated carbon sink enhancement per unit of deposited nitrogen is 3.7 g C/g N, which is slightly lower than the range of 4.3-4.8 g C/g N by previous global simulations.

3. It would be interesting to have at least a short description on the sub-annual characteristics of the LAI changes. Are you seeing changes in the peak, or just a lengthening of the growing season? can you say which is more important at driving a carbon sink? There is much literature on this and the offset of increased respiration too. The recent review by Piao et al (GCB, 2019) has a lot of detail. Also, e.g. Buermann et al (Nature, 2018).

This is a useful point to add. According to our analysis, we found that the mean LAI increase as a global average is 0.1 over the 1981-2016 period, while the corresponding increase in the maximum LAI is 0.2, suggesting that the overall LAI increase is more due to changes in the peak than the lengthening in the growing season. However, both increases in the mean and maximum LAI depend on latitude (Figure A2). Tropical areas show increases in the maximum LAI and little changes in the mean LAI while temperate and boreal areas show increases in both,

although increases in the maximum LAI are generally larger than those in the mean LAI. These results mean that increases in the peak LAI is the main reason in the global LAI increase, while the lengthening of the growing season could also be a reason especially at latitudes of 55-65° N (boxed in Figure A2), in broad agreement with the conclusion of Piao et al (2019) that both the start of the growing season has shifted early by 2-4 days/decade and the end of the growing season shifted later by 2-3 days/decade on average. Buermann et al (2018) claimed that about 15% of northern ecosystems showed declines in the summer productivity as compensation for longer growing seasons. This is also reflected in our LAI products, which show that 39% of vegetated areas in 55-65° N latitudes had decreased maximum LAI in the summer over the 1981-2016 period, while the remaining 61% showed the opposite trend. Since the seasonal pattern is an important point, we added in the main text (Lines 111-113) the following sentence: *“Globally, the increase in the maximum LAI in the peak growing season is about twice as large as that in the annual mean, suggesting that growing season lengthening is not the main reason.”*

Figure A2. Increases on the mean and maximum LAI at different latitudes over the 1981-2016 period, showing widespread greater increases in the maximum than in the mean LAI, suggesting that LAI increases more in the peak growing season than in other seasons.

4. in places I felt the methods could be described in more detail in the main text. I don't know how to recommend how to achieve this within word count, but just to suggest you describe the spin-up process. After reading the main text I was fairly skeptical about the nature of how you represent the sink prior to and after 1981. But when I read the supplement this became clear and I am happy with the method. But a reader might also query this - I would suggest at least a mention that you initialise your carbon pools with a pre-industrial spin up and then run up to 1981 before starting your factorial simulations from there.

Following your suggestion, we modified and expanded the following text (Lines 292-298):

“The initial sizes of these carbon pools at 1901 for each modeling grid are estimated through solving a set of linear equations for decomposition and transfer between the pools under the

dynamic equilibrium assumption which equates NPP at 1901 with HR³⁰. Historical climate, CO₂ and nitrogen deposition data are then used to model changes of plant growth and carbon pools from 1901 to 1981 in preparation for detailed modeling for the 1981-2016 period.”

5. related to this I question your assertion that you don't include any LAI changes during the period up to 1981. Given that CO₂ has increased by about the same amount before/after 1981 (approx. 280-340 and then 340-400 ppm), then naively I might expect the LAI change up to 1981 is similar magnitude to the change after it. Have you looked at the sensitivity of your results to the assumption it did not change? What if you extrapolated back to pre-industrial a LAI change of equal magnitude to the changes after 1981? How would this affect your legacy sink?

This is an interesting question, which is a logical extension from our current work. We agree that it is also an important issue to be addressed, and it may deserve further study. However, the focus of our current paper is on the satellite era with many sensors tracking the actual changes in vegetation. As reliable LAI data prior to 1981 are not yet available, we are not able to extend the same modeling to earlier dates. Any vegetation change in the pre-satellite era would have little effect on GPP estimation in the satellite era as GPP is based on daily LAI. However, heterotrophic respiration would be affected if the LAI changed substantially prior to 1981. For this purpose, we conducted a new set of simulations by extending the LAI time series to 1901 according to atmospheric CO₂ concentration with the rate of LAI change determined using 1981-2016 data. We found that the enhancement of the land sink due to LAI change in 1981-2016 decreased from 12.4% to 10.5% relative to the accumulated sink in the same period when the possible LAI increase from 1901 to 1981 is considered. This decrease is due to higher total accumulated NEP over 1981-2016 by 6.8 Pg C resulting from lower initial soil carbon pools at 1901 when LAI is smaller than our previous simulations without considering LAI change over 1901-1981. This new set of simulations suggests that the impact of possible LAI changes prior to 1981 on LAI effects after 1981 is within a few percent and does not affect our conclusion on the significance of LAI increase after 1981 in enhancing the land sink. Although extrapolating LAI according to CO₂ concentration is overly simplistic, it may be considered as setting the upper bound of the possible error due to LAI changes prior to 1981 because climate change could have been negative on plant growth and LAI. As it needs considerable space to make the above point, we choose to discuss this in the supplementary material (Lines 35-47)

Minor comments

1. Abstract. line 47. Can you explain what you mean by "positive" feedback? Do you mean positive as in "good" in a subjective sense? or positive as in "an amplifying feedback"? If the latter, then I don't understand - you show that LAI changes have caused a carbon sink - surely this is a negative feedback? **We mean “an amplifying feedback” on the carbon sink, i.e. environmental factors (including CO₂, N and climate) cause increase in GPP and therefore in LAI, which in turn further increases GPP and carbon sink. We now use “an amplifying feedback” on land sinks in Line 47 to avoid confusion of positive or negative feedback which generally refers to impacts on temperature.**

2. you describe your model as "diagnostic" - I'm not sure what this means exactly. I would say at least parts of it are prognostic. It is driven by observations and forced by the observed LAI - that's fine. But

in the sense that it has a memory (i.e. "state variables") which evolve from timestep to timestep, then it is a prognostic model. Certainly, the soil component, if based on CENTURY, is very similar in structure to the other process based prognostic models you describe.

It is indeed that our GPP model is diagnostic while our soil model is not entirely diagnostic because the changes in the soil carbon pools are tracked in a similar way like prognostic models. However, these changes are also affected by LAI changes, and therefore we would simply characterize our entire model as diagnostic.

3. lines 233/234. You don't need to show working out to calculate a percentage.

Agreed, it is removed.

4. SI section 2.2. Can you briefly describe how this N-dep data set compares with that used within CMIP6 and TRENDY for driving process models?

The N-dep dataset is now described in supplementary material (Lines 112-120), and Figure S2 (or Figure A3 below) is added to compare the N deposition datasets, as the global average, used in our study and in TRENDY models. The two datasets are similar before 1990 as both are based on measurements, but the increasing trends after 1990 are different because the data used by the models are based on linear extrapolation from 1990 to 2050, at which the N deposition is estimated based on projected anthropogenic sources (Dentener, 2006), while our dataset is based on satellite measurements for the 2000-2009 period (Lu et al., 2014) and could be more realistic than the linearly extrapolated trend used by the models.

Figure A3. Comparison of N deposition datasets used in this study and by TRENDY models

5. Suppl. figures 3, 4 etc. It would be useful to be able to relate the threshold inland-use of 20/30% to your x-axis here, which is expressed in % per year. Perhaps you could mark a dashed line upwards from the points you include/exclude due to land-use

Thanks for the suggestion. We have now added the suggested lines to the figures (now Figures S4, S5 and S6)

6. Suppl. figures 3-5. Please can you use a common colour scale with zero in the same place? In each of these figures, the zero point is different which means sometimes yellow is a positive value, sometimes negative etc. It would make them more readable as a set if the same colours represented positive/negative changes.

Thanks for the suggestion. Two of the figures (now Figures S5 and S6) with positive and negative values now have the same color code for the zero point.

7. Suppl. methods lines 193-196 - please check figure numbering here. Do you really mean figures 5 and 6?

We have checked the numbering sequence and they match up correctly now (Lines 228-237 now in supplementary material).

8. Suppl. figures 7 onwards - the maps of different runs. Please check the caption definition of which simulations you show and check against the table listing the runs. E.g. fig S7 says "Simulation VI" but I think it is "II" in the table? S8 says "II" but means "III"?

The simulation schemes are now mentioned in figure captions correctly.

9. these maps are really nice - please can you add the final one showing the committed sink just from the changes up to 1981. You conclude that this is about 60% of the total carbon sink, so it would be nice to see the map too.

This is a good idea. We have added a figure showing the legacy (committed sink) effect on the sink in the 1981-2016 period (Figure S13).

Reviewer #3 (Remarks to the Author):

This paper addressed an important question regarding the impact of long term vegetation changes on the Terrestrial Carbon Sink, using remote sensing data from 1981 to 2016. To my knowledge, the work is original. I judged the article and supplementary material well-structured and scientifically high level. I have some comments, mostly related to vegetation remote sensing, my domain of expertise.

Thanks for these positive comments.

Major comments:

The long term trend of LAI have been studied by Jiang et al. (2017). The current article relies on these previous study to analyze the impact of LAI trend to GPP and carbon sink. Jiang et al. clearly concluded (section 4.3) by: "A single product or even ensemble of the four products may not be directly considered as relevant references when interpreting long-term global carbon and water cycle studies during pre-MODIS period (1982 – 1999), except for several consensus: (i) trend signs, (ii) interannual variability values in high latitude regions, and (iii) annual anomalies in arid regions.". However, the current article is based on the trend on LAI, not only the sign (see "(i)" above) but also the value of the trend I think the authors should demonstrate the validity of using trend values knowing the conclusion of Jiang et al.

Thanks for raising this important issue about LAI trend. In addition to the detailed answer to

the same question by Reviewer 1 (see above), we would like to say that research on LAI mapping has advanced considerably since Jiang's paper. As shown in supplementary material (Figure S1 and Table S1), all LAI products show highly significant trends, and the uncertainty on the trend and its impact on sink estimation are included in Figure 2.

Can we trust the LAI variation, especially when we know the poor quality of the LAI products for pure-AVHRR-era (<2000)? I understand that the attribution of LAI to global land sink (figure 4) is determined by the analysis of Simulation III vs Simulation I. Therefore authors compare Simulation with LAI from 1982-86 (Simulation I) and Simulation with actual Changing LAI values (1981-2016) (Simulation III). However, the Supplementary Figure 1 (from Jiang et al. 2017 – original copied below) highlights the fact that there is a significant change for all product from 1999 (pure-AVHRR-era) to 2003 (MODIS-era). Therefore what is the validity of the comparison between Simulation I and Simulation III?

We appreciate this insightful and pointed question. Indeed, the impact LAI on the land sink depends on both the trend and the reference point set as the average of 1982-1986 values. If a product with a large deviation of LAI values in this 5-year period from the trend line, large uncertainty would occur. However, all three products show a good fit of the trend line to the data points in these five years (Figure A1 above or S1 in supplementary material), and therefore the uncertainty due to this reference point relative to the LAI trend is very small and can be ignored (Figure 2). We would like to emphasize that the magnitude of the sink enhancement due to LAI increase over the 1981-2016 period depends mostly on the LAI trend, while its absolute value and interannual variation have only secondary effects.

[please insert attachment_1 here]

The trend is derived from one LAI product and two more are analyzed in supplementary material. I suggest to better argue why they choose GLOMAP product as core, and GLASS and GIMMS-LAI3g, as secondary. Why not using all products covering the entire period? Indeed Jiang et al. concluded that TCDR was the most inconsistent product after GLOMAP. Why removing TCDR from the analysis in the supplementary material?

The TCDR product uses the same AVHRR data as GLASS and is very similar to the new GLASS product (revised after Jiang's paper) (Figure S1). Using both of them would not help much our uncertainty assessment, and therefore we used GLASS and not TCDR.

Minor comments:

Figure 1: describe the lower left graph. In order to improve readability of the graph, I suggest to better identify "increase" and "decrease" in the xlabel.

We have added "increase" and "decrease" on the color bar.

Figure 1 (and many other places): what does the unit "a-1" refer to? Is there a confusion with year (yr-1)? If so homogenize.

We changed "a-1" to "yr-1".

L. 174: for more clarity, I suggest to add, like in supplementary (section 4), "Disturbance (i. e. land

cover change)"

Similar explanation in brackets is already given just one sentence prior to this line (now Line 182).

Figure 4: Not very clear - I understand that positive and negative values were separated to draw the cumulative bar lines. I would suggest to better distinguish positive and negative values (see graphical suggestion)

We agree. The negative values are now given in shaded areas.

[please insert attachment_2 here]

Supplementary Figures 7-10: these figures are informative and map of results would be appropriate at the end of the core article. However, I understand that the 4 full map could not be placed in the article. I would suggest several options (but authors may think to better ideas):

- add fig 8 in article since LAI is the main variable of interest
- add maps of relative contribution of dNEP (but authors have to deal with positive vs negative contributions)
- use a RGB color composite by mapping a set of three most important contributions

Many thanks for offering these excellent suggestions to better present our results. Among the three options, we choose the third as it provides an effective summary of the impacts of the four main drivers (CO₂, LAI, N deposition and climate) on the accumulated sink enhancement over 1981-2016 (Figure S14). As shown in Figure S14, climate had the most dominant negative impact on the accumulated carbon sink in 14.2% of the global total vegetated area. LAI is the dominant positive driver for 43.6% of the area, while it is negative for 4.6% of the area. CO₂ and nitrogen deposition are both positive dominant factors over 36.4% and 0.2% of the area, respectively. This is added to the supplementary material (Lines 286-289).

Typographical mistakes:

- L.84: "total lank sink" => "total land sink"

We see no need to change the phrase. The first "the land sink" has a slightly differing meaning from the second "the total land sink". So they are unchanged.

- Supplementary Table 1: Row3, Col 1: "I" => "II"

Changed. Now it is Table S2.

REVIEWERS' COMMENTS:

Reviewer #2 (Remarks to the Author):

I thank the authors for their positive responses to my comments. I hope I helped you get your thinking straight about how to present the results. I have no remaining comments to add and am happy this can be accepted for publication.

Reviewer #3 (Remarks to the Author):

The authors correctly answered my main doubt related to the reality of the trend observed on LAI RS products. All products show a positive LAI trend and it is true that many efforts from the RS scientific community have contributed to have well calibrated TS. This is well discussed in the response to Reviewer #1.

Therefore, I consider the manuscript good for publication.

Just one minor question: if not typo error, what does "lank" mean in the expression "total lank sink" (L. 84) and "the total residual lank sink" (L. 274)?